# Recent Advances in Organic Thermoelectric Materials: Principle Mechanisms and Emerging Carbon-Based Green Energy Materials

**DOI:** 10.3390/polym11010167

**Published:** 2019-01-18

**Authors:** Yinhang Zhang, Young-Jung Heo, Mira Park, Soo-Jin Park

**Affiliations:** 1Department of Chemistry, Inha University, 100 Inharo, Incheon 22212, Korea; bank0719@163.com (Y.Z.); heoyj1211@inhaian.net (Y.-J.H.); 2Department of Bioenvironmental Chemistry, College of Agriculture & Life Science, Chonbuk National University, Jeonju 54896, Korea

**Keywords:** graphene, CNT, organic thermoelectric materials, Seebeck coefficient

## Abstract

Thermoelectric devices have recently attracted considerable interest owing to their unique ability of converting heat to electrical energy in an environmentally efficient manner. These devices are promising as alternative power generators for harvesting electrical energy compared to conventional batteries. Inorganic crystalline semiconductors have dominated the thermoelectric material fields; however, their application has been restricted by their intrinsic high toxicity, fragility, and high cost. In contrast, organic thermoelectric materials with low cost, low thermal conductivity, easy processing, and good flexibility are more suitable for fabricating thermoelectric devices. In this review, we briefly introduce the parameters affecting the thermoelectric performance and summarize the most recently developed carbon-material-based organic thermoelectric composites along with their preparation technologies, thermoelectric performance, and future applications. In addition, the p- and n-type carbon nanotube conversion and existing challenges are discussed. This review can help researchers in elucidating the recent studies on carbon-based organic thermoelectric materials, thus inspiring them to develop more efficient thermoelectric devices.

## 1. Introduction

The thermoelectric effect, which can directly convert a temperature difference to electricity within a conducting material, was discovered in 1821 by Thomas Johann. Soon after, the development of efficient thermoelectric materials and devices widely emerged in the industrial and scientific communities [1,2,3,4,5,6]. In comparison with other energy transition materials, thermoelectric materials are endowed with various advantages, such as direct energy conversion without a moving part or working fluids, no sound or gas pollution, no working position restrictions, no scale effect, long working lifespan, and wide applications in aerospace, cogeneration, and wearable devices [7,8,9,10,11,12,13].

Inorganic semiconductors, such as SnSe [14,15,16], Bi_2_Te_3_ [17,18,19,20,21], SiGe [22,23,24], SnS [25,26,27], and CoSb_3_ [28], exhibit excellent thermoelectric performance, and some of them have been commercialized with the development of the temperature difference generator and semiconductor temperature difference refrigerator. However, their applications are limited owing to various disadvantages, such as their high cost, relatively heavy weight, strong toxicity, and processing difficulty. Conversely, organic polymer-based thermoelectric materials are promising as a new generation of thermoelectric candidates, owing to their unique merits, such as light weight, mechanical flexibility, non-toxicity, easy processing, and inherently low thermal conductivity [29,30,31,32,33,34,35]. In the early days of their development, these organic thermoelectric materials did not have any practical significance owing to their small thermoelectric effect. However, great progress has been recently achieved, and excellent thermoelectric performance similar to that of conventional inorganic thermoelectric materials has been observed. Polymers employed in organic thermoelectric materials are categorized into conducting polymers and non-conducting polymers. The commonly investigated conducting polymers include poly(3,4-ethylenedioxythiophene) [36,37,38,39], polyacetylene [40,41], poly(aniline) [42,43], polythiophenes [44,45], polypyrrole [46,47], polyphenylenevinylene [48], and poly(3-methylthiophene) [49,50], while the non-conducting polymers include poly(3-octylthiophene) [51], poly(3-hexylthiophene) (P3HT) [52], and polyvinylidene fluoride [53,54,55]. Their chemical structures are presented in Table 1. Owing to their intrinsically electrical conductivity, the conducting polymers are preferable and cover the majority of the thermoelectric device matrix. In non-conducting polymer-based composites, non-conductive polymers act as a barrier to bundle-to-bundle hopping, which decreases their thermoelectric performance [52,56,57]. Nonetheless, the further development of organic thermoelectric devices has been hindered because polymer materials have a low electrical carrier, Seebeck coefficient, and power factor. Moreover, the power factor of organic thermoelectric materials is three orders of magnitude lower than that of conventional inorganic thermoelectric materials [58,59,60]. To overcome these restrictions, organic polymer-inorganic thermoelectric composite structures have been developed, and satisfactory thermoelectric performance has been obtained over time.

In this review, we briefly summarize the mechanisms and parameters that can affect the thermoelectric performance, and focus on summarizing recently developed carbon material-based organic thermoelectric composites. The carbon nanotube (CNT) and graphene are the most advanced carbon materials endowed with remarkable electrical conductivity, and can saliently increase the electrical conductivity of composites. Carbon-based thermoelectric materials are characterized by the decoupling of electrical conductivity and Seebeck coefficient, and their thermoelectric performance is promising [61,62,63]. Moreover, prepared composites have a low thermal conductivity owing to phonon scattering, which occurs at the carbon/polymer interfaces and makes carbon-based organic thermoelectric materials more efficient and commercially feasible, owing to their high reliability and environmental efficiency [64].

## 2. Principle Mechanisms of Thermoelectric Materials

Figure 1a shows a macroscopic illustration of how heat can be converted to electricity (Seebeck coefficient) using thermoelectric materials. Figure 1b shows how electrical power can be transformed to cooling or heating (Peltier effect). In brief, when temperature differences occur at the two ends of a thermoelectric material, the mobility of the hole and/or electrons is promoted, which results in current flows (Figure 1a). When electricity is applied to thermoelectric materials, holes and/or electrons are generated and forced to flow at the two ends. Thus, the one end cools down while the other end becomes heated (Figure 1b). Thermoelectric devices are assemblies of n-type (electron rich) and p-type (electron deficient) semiconductors. The thermoelectric energy conversion efficiency can be evaluated using the dimensionless parameter known as the figure of merit (*ZT*) [57,65,66], as follows:(1)$$ZT=\frac{\sigma S^{2}T}{k}$$
where σ, *S*, *k*, and *T* are the electrical conductivity, Seebeck coefficient, thermal conductivity, and absolute temperature, respectively. It can be seen that, at a specific temperature, the high Seebeck coefficient, large electrical conductivity, and low thermal conductivity are the key factors for achieving excellent thermoelectric materials. However, it has been found that these parameters are strongly interconnected and interdependent, which leads to extreme difficulty in optimizing and improving the thermoelectric efficiency (*ZT*) in bulky materials [67,68]. Owing to the intrinsically low thermal conductivity of polymer-based organic thermoelectric devices, their thermoelectric efficiency is evaluated using a new criterion known as the power factor (*PF* = *S*^2^σ) [69,70]. The compromise and synergy of these parameters in determining the thermoelectric performance will be elaborated in the following sections.

### 2.1. Electrical Conductivity

According to Equation (1), the thermoelectric efficiency is positively correlated with the electrical conductivity of thermoelectric materials. The electrical conductivity is typically expressed through Equation (2), as follows:(2)σ=μqn
where µ is the electron mobility, *q* is the elementary charge, and *n* is the charge carrier density. Moreover, *n* is related with the doping level of conductive polymers, and µ is affected by the chemical morphology and structure [71,72]. Equation (2) does not only apply to materials when the electrical conductivity results entirely from the electron flow (n-type semiconductor), but also applies to p-type materials, whose electrical conductivity is attributed to the hole flow. In the case wherein a semiconductor contains electrons and holes simultaneously, the electrical conductivity is the sum of the p-type and n-type conductivity. The electron or hole mobility depends on the drift velocity [73,74], which is determined by the scattering time. Ionized impurity scattering and acoustic phonon scattering are two of the most crucial scattering sources in thermoelectric materials [75]. Ionized impurity scattering is the scattering of charge carriers caused by ionization occurring in the lattice, while acoustic phonon scattering is the phonon state change that occurs during the collision of phonons in a non-linear interaction. Charge carrier density is denoted as the number of charge carriers per volume, and is a crucial quantity in the process of chemical doping.

### 2.2. Seebeck Coefficient

The Seebeck coefficient is defined as the electric voltage (Δ*V*) generated when a small temperature gradient (Δ*T*) is applied to a specific material, and is expressed by Equation (3) [76,77], as follows:(3)$$S=\frac{\Delta V}{\Delta T}$$

The Seebeck coefficient of a material varies with the changes in the external temperature, and depends on the chemical composition of the materials at a specific temperature. In quantum theory, the Seebeck coefficient is a measure of entropy for a carrier with unit charge [78]. In a simple charge carrier system with low interactions, the Seebeck coefficient can be determined using Equation (4) [72], as follows:(4)$$qS=\frac{k_{B}\ln\left[\left(1-\rho \right)/\rho \right]}{\Delta T}$$
where *k_B_* is the Boltzmann constant and ρ is the charge density inside the materials. Apparently, as the charge density increases, the electrical conductivity increases accordingly, while the Seebeck coefficient decreases, which indicates an interdependent relationship between the Seebeck coefficient and the electrical conductivity. Consequently, there should be an optimal doping level to obtain the highest *PF* value.

### 2.3. Thermal Conductivity

Thermal conductivity is an intrinsic property of materials, and can be evaluated using Fourier’s law [79,80]. The standard unit for thermal conductivity is watts per meter-kelvin (W/(m·K)); its reciprocal is defined as the thermal resistivity and is expressed in Kelvin-meters per watt (K·m·W^−1^) [81]. Materials with higher thermal conductivity are typically used as heat sinks, while materials with lower thermal conductivity are typically used for thermal insulation. In the case of thermoelectric materials, lower thermal conductivity is beneficial for obtaining a higher thermoelectric efficiency according to Equation (1). Therefore, clarifying the mechanism of thermal conductivity is very important for the development of thermoelectric materials. In quantum theory, thermal conductivity is the ability of transferring the vibrational energy from one atom to its neighboring atom so as to reach thermal equilibrium by collision without shifting the matter [82,83,84]. The thermal conductivity (*k*) is realized by the phonon synergy (*k*_L_) and electron transport (*k*_e_). The phonon transfer plays a crucial role in the thermal conductivity of carbon and polymer-based materials [85]. A phonon is a quantized lattice vibrational energy that can realize heat transfer in materials through lattice vibration [86,87]. In brief, the thermal conductivity of a material is the process of phonon transfer. The phonon transfer mechanisms in pure polymer composites are shown in Figure 2, where it can be seen that the phonon can be directly transmitted without any scattering when phonon transfer occurs between the same atoms. However, phonon scattering occurs when phonons are transferred between different atoms, and when defects, impurities, and an amorphous phase exist in the interface between the adjacent atoms. Typically, polymers have very low crystallinity, and the interfacial region amongst the different atoms is amorphous. This promotes the formation of many defects and discontinuous internal interfaces and results in the enhancement of phonon scattering and the deterioration of heat diffusion efficiency or phonon transfer efficiency [88,89]. Therefore, a pure polymer has an intrinsically lower thermal conductivity in comparison with crystalline inorganic materials. The thermal conductivity mechanism of inorganic materials is very different from that of the abovementioned polymer composites. Phonon scattering, which can hinder the transfer of phonons in a straight wave, is widespread inside polymers owing to their irregular arrangement of atoms. However, in crystalline particles, the phonon energy can vibrate with a frequency of ν = ε/*h* to adjacent atoms and realize heat conduction from hot to cold surfaces. Here, *h* is Planck’s constant, ν is the vibration frequency and ε is the vibration energy or energy absorbed from the surroundings [90,91]. The proposed mechanism of thermal conductivity is shown in Figure 3. The phonon can be directly transferred amongst atoms without much scattering. Slight phonon scatterings inevitably occur when phonons transfer through surface defects. The thermal conductivity of polymer-based thermoelectric composites is the combination of the thermal conductivity of inorganic crystalline particles with that of organic polymers, but it does not emerge simply by adding them up. Calculating and predicting the thermal conductivity of thermoelectric polymer composites is highly complex because it is a function of the filler structure, filler network, intrinsic thermal conductivity of the filler and polymer, filler dispersion, and filler interfacial thermal resistance. Owing to the discontinuous network shown in Figure 4, phonon scattering caused by acoustic mismatch occurs in different atoms and is widespread at the polymer matrix/filler interfaces. This results in the lower thermal conductivity of polymer composites, which is beneficial to the thermoelectric efficiency [92].

## 3. Carbon-Based Organic Thermoelectric Materials

Carbon materials, such as graphene, CNT, nanodiamonds, carbon fiber, and fullerenes, have attracted great interest in recent years [93,94,95,96,97,98,99,100,101,102,103,104,105,106,107]. In the past decades, carbon materials, particularly CNT and graphene, have been widely employed in the fabrication of thermoelectric materials because of the following reasons: (1) carbon materials have an intrinsically high electrical conductivity, which can significantly enhance the thermoelectric efficiency of thermoelectric materials; (2) as novel carbon nanomaterials, their large specific surface areas can promote the formation of a highly efficient interface between the polymer matrix and the carbon particles [93,108,109,110,111,112,113,114]; (3) the high thermal conductivity of carbon materials can be alleviated by wrapping or coating the polymer matrix on their surfaces [115]; and (4) carbon-based thermoelectric polymer composites are flexible, low-cost, and non-toxic, in addition to having high mechanical strength and being light-weight. However, their thermoelectric performance is inferior to that of conventional inorganic thermoelectric materials. More extensive investigation is needed for developing highly efficient polymer-based thermoelectric materials that can be made commercially available.

### 3.1. CNT-Based Organic Thermoelectric Materials

CNTs with a unique one-dimensional structure have attracted a great amount of interest with regard to the fabrication of thermoelectric materials because, theoretically, one-dimensional nanomaterials exhibit better thermoelectric performance in comparison with multi-dimensional materials [116,117,118,119]. The thermoelectric efficiency can dramatically increase by enhancing the mobility along the long direction of the CNTs, or by lowering their diameters to decrease the thermal conductivity. CNTs have a very high mobility of 100,000 cm^2^/(V·s) [120] and a thermal conductivity of 3000 W·m^−1^·K^−1^ [121], while organic polymers have a carrier mobility lower than 50 cm^2^/(V·s) [120] and a thermal conductivity of approximately 0.2 W·m^−1^·K^−1^ [122]. Optimizing the synergy and compromising the electrical conductivity and thermal conductivity are key factors in the fabrication of highly efficient thermoelectric materials.

CNT is often regarded as a typical p-type thermoelectric material because holes are the primary charge carriers for the CNT. The opposite was thought to be true until 2000, when Bradley found that CNT exhibits an n-type thermoelectric property under a vacuum environment, which suggests that pristine CNT is intrinsically an n-type material [123,124]. Their study demonstrated that the Seebeck coefficient is sensitive to the concentration of ambient oxygen, which can be used to convert a semiconducting CNT into a metal through exposure [123]. In an atmospheric environment with different dopants, semiconducting CNT can also exhibit n-type thermoelectric performance, while the flexible p-type to n-type transition expands its potential applications and research scope in industrial and scientific fields. A summary of thermoelectric results of CNT-based organic thermoelectric materials is presented in Table 2.

#### 3.1.1. p-Type CNT-Based Organic Thermoelectric Materials

The p-type and n-type thermoelectric sections are two essential parts of an integrated thermoelectric device. In most cases, CNT has been investigated as a p-type thermoelectric material and doped by an oxidative dopant, such as chemical oxidants, oxygen, and acids. Various recent p-type CNT-based thermoelectric polymer composites are summarized in this section, including the doping type, advanced composite fabrication methods, and achievements of related studies.

Wang et al. [136] prepared Polyaniline/MWCNT (PANi) thermoelectric polymer composite nanofibers using in situ polymerization and electro-spinning processes. The particular design employed in this study consisted of both CNT and polymer nanofibers aligned and oriented along the same axis. The highly arranged PANi backbone chains did not only decrease the π–π conjugated defects amongst the polymer molecules as a result of ring twisting, but also enhanced the carrier mobility by enhancing the effective degree of electron delocalization. As expected, the PANi/MWCNT polymer composites with highly-arranged nanowires exhibited superior anisotropic thermoelectric performance. The electrical conductivity and power factors were two times higher in the parallel direction in comparison with those of the directly mixed PANi/MWCNT polymer composites and those of the PANi/MWCNT polymer composites arranged in the perpendicular axis.

Meng et al. [126] developed a facile strategy for synthesizing high-performing CNT-based thermoelectric polymer composites. In brief, a freestanding network containing CNT bundles was obtained by filtering the CNT suspensions with the assistance of a vacuum. Then, the fabricated CNT networks were coated with PANi layers synthesized by an in situ chemical polymerization method. The results revealed that the PANi/CNT sheet composites exhibited a much higher Seebeck coefficient of 22.4 μV·K^−1^ with regard to the neat CNT (12.2 μV·K^−1^) and pure PANi (2.74 μV·K^−1^) at 300 K. The enhanced Seebeck coefficient is attributed to the size-dependent energy-filtering effect caused by the PANi wrapping. Moreover, the PANi/CNT pellet and PANi/array nanocomposites were fabricated and exhibited a Seebeck coefficient of 23.5 and 25.5 μV·K^−1^, respectively, which demonstrates that the PANi coating is a promising and effective approach for enhancing the performance of thermoelectric materials.

Gao et al. [137] proposed a novel synthesis strategy for preparing flexible films of thermoelectric materials by oxidizing thieno[3,4-*b*]pyrazine (TP) into TP di-N-oxide (TPNO) on SWCNT surfaces using an in-situ oxidation reaction method. The thermoelectric performance of the prepared materials was significantly enhanced and could be tuned by the content of poly(sodium-*p*-styrenesulfonate), mass ratio of TPNO to SWCNT, and various post-treatments using polar solvents. The electrical conductivity of the prepared composites was monotonously enhanced from 2.0 to 382.8 S·cm^−1^ with the reduction of the TPNO/SWCNT mass ratio, which is attributed to the increase of carrier mobility caused by the increased CNT concentration. The Seebeck coefficient decreased with the increase of the SWCNT concentration, and the highest power factor reached up to 29.4 μW·m^−1^·K^−2^ when the TPNO/SWCNT ratio was 2:1. All samples subjected to post-treatment exhibited superior thermoelectric performance in comparison with the untreated samples. The samples treated with DMSO had the highest electrical conductivity of 185.2 S·cm^−1^ and highest power factor of 17 μW·m^−1^·K^−2^, which is twice as much as that of untreated films. The enhanced electrical conductivity is attributed to the hydrophilic solvents with a high boiling point, which facilitated the dissolving of polymer chains and the formation of interconnected SWCNT networks. Additionally, this demonstrates that the Seebeck coefficient, which depends on the entropy per charge carrier [138], can be tuned by carrying out post-treatment with polar solvents.

Hong et al. [125] prepared high-performing thermoelectric P3HT/CNT film and P3HT/CNT organic thermoelectric devices using the spray-printing method. The P3HT used herein was not doped and exhibited a low electrical conductivity of 10^−5^ S·cm^−1^. However, the strong π–π interaction between the P3HT and the CNTs improved the dispersion of the CNT. As a comparison, P3HT/CNT organic thermoelectric materials were also prepared using the conventional drop-casting method. The thermoelectric materials prepared using the spray-printing method exhibited a Seebeck coefficient of 97 μV·K^−1^, electrical conductivity of 345 S·cm^−1^, and a power factor of 325 μW·m^−1^·K^−2^, while the thermoelectric materials prepared using the drop-casting method exhibited a Seebeck coefficient of 102 μV·K^−1^, electrical conductivity of 224 S·cm^−1^, and power factor of 234 μW·m^−1^·K^−2^ at room temperature. The enhanced electrical conductivity of the spray-printed thermoelectric materials is attributed to the higher density of the inter-CNT bundle connections. A flexible organic thermoelectric device composed of spray-printed P3HT/CNT composites was prepared. The device had an open-circuit voltage of 41.8 mV and maximum power factor of 32.4 nW. This study demonstrated that the spray-printing method is a promising approach toward fabricating flexible and low-cost thermoelectric materials

The vertically-aligned MWCNT forest (VA-CNTF) as a filler for conductive polymers was first investigated by Yusupoy et al. [139] to enhance the thermoelectric performance of materials. PEDOT:PSS was selected as a dispersion medium owing to its superior electric conductivity. In comparison with pristine CNT, the VA-CNTF-based materials have a distinctly increased power factor. Without subjecting the VA-CNTF sample to pre- or post-treatment, a dramatically increased Seebeck coefficient of 46 μV·K^−1^ and the highest power factor of 56.5 μW·m^−1^·K^−2^ were obtained for a one-layer sample at 420 K. The enhanced thermoelectric properties are attributed to the bundles of vertically-oriented MWCNT, which induced the analogical alignment of the PEDOT molecule chains.

Lee et al. [130] fabricated a DWCNT/PEDOT:PSS nanocomposite film, which was subsequently subjected to post-treatment in ethylene glycol (EG) for 1 h and annealing at 140 °C for 10 min. The EG treatment mostly removed the non-complex PSS molecule chains, and thus effectively decreased the inter-DWCNT and inter-bundle distances in the composite films. This novel method dramatically enhanced the electrical conductivity of the composites. With EG treatment, the CNT/PEDOT:PSS nanocomposite film with 30% CNT exhibited an electrical conductivity of 736 S·cm^−1^, which was approximately five times higher than that without EG treatment. The DWCNT/PEDOT:PSS nanocomposite film with 20% DWCNT had the highest power factor of 151 μW·m^−1^·K^−2^, while the same sample without EG treatment had a considerably inferior power factor of 21.6 μW·m^−1^·K^−2^. The EG post-treatment provided a promising, cost-effective, and easy-processing method of preparing high-performance thermoelectric materials. An organic thermoelectric device composed of a DWCNT/PEDOT:PSS nanocomposite film and silver electrode line achieved a maximum power output of 30 Nw, which demonstrates its capability of generating a substantial amount of electric power.

Hsu et al. [140] prepared PEDOT:PSS/CNT thermoelectric materials with an unchanged or increased Seebeck coefficient, when the electrical conductivity increased with the increase of the CNT concentration or by post-treating the materials in dimethyl sulfoxide (DMSO) or formic acid (FA). Note that this is contrary to the correlation of electrical conductivity with the Seebeck coefficient. The electrical conductivity of the PEDOT:PSS/CNT sample that contained 20% CNT and was treated by DMSO or FA was remarkably enhanced to 1.8 × 10^3^ S·cm^−1^ from 0.31 S·cm^−1^ for the pristine PEDOT:PSS composite. The well-percolated and electrically connected CNT networks, which can facilitate the carrier transport, were responsible for the high electrical conductivity. The Seebeck coefficient of the PEDOT:PSS/CNT composites reached a maximum of 59 μV·K^−1^, when the CNT concentration reached 6.7 wt %, and then monotonically decreased as the CNT concentration increased. Large power factors of 464 and 407 μWm^−1^K^−2^ were obtained for the composites containing 6.7 wt % CNT after DMSO and FA treatment, which outperforms most of the results obtained by this study with the same CNT loading.

#### 3.1.2. n-Type CNT-Based Organic Thermoelectric Materials

It should be noted that the n-type CNT is much more challenging and complex to prepare in comparison with p-type CNT, because most CNT behaves as p-type in air, owing to the oxygen. Strategies of converting the p-type CNT to n-type CNT include chemical reduction and low molecular doping. However, the reduced CNT is unstable under ambient conditions owing to the existence of oxygen. Consequently, electron-donating dopants are widely used to realize the p-type to n-type conversion. The p- to n-type transition is the process of switching the carrier type majority, where the holes and electrons correspond to the p-type and n-type conduction behaviors, respectively. Pristine CNT exhibits the p-type behavior because holes are the dominant charge carriers. After doping treatment, the Fermi level shifts upwards to the conduction band of the CNT as a result of the charge transfer interaction between the CNT and the dopants. Consequently, the Fermi level shifts closer to the conduction band, which realizes the conversion from p- to n-type CNT and, eventually, the reduction of electrical conductivity [141].

David et al. [142] have proposed a novel spray doping method of preparing carbon nanotube-based polymer composites with easily tunable thermoelectric performance. Initially, the p-type thermoelectric composites of CNT and poly(vinylidene fluoride) (PVDF) were fabricated using a drop casting method. Then, small molecule n-type dopant polyethyleneimine (PEI) was doped into the film with a three-dimensional (3D) printed thermoplastic stencil mask to realize the conversion from p-type to n-type. The results revealed that a fast transition from p-type to n-type occurred at the PEI to CNT mass ratio of 0–0.2, and that the Seebeck coefficient reached the maximum value of −32.5 μV·K^−1^. The electrical conductivity increased as the CNT loading increased owing to the percolating networks, and then decreased owing to the Fermi level transition.

A novel class of high-performance hybrid organic thermoelectric materials was first developed by Toshima et al. using advanced nanotechnology and without employing conducting polymers such as (poly(3,4-ethylenedioxythiophene)) [143]. The hybrid thermoelectric material consisted of SWCNT, poly(vinyl chloride) (PVC), and nanoparticles of the n-type semiconducting polymer complex poly(nickel 1,1,2,2-ethenetetrathiolate) (PETT). The prepared three-component films had a Seebeck coefficient of 30.2 μV·K^−1^, and a thermoelectric power factor of 39.117 μW·m^−1^·K^−2^. Moreover, this study found that the electrical conductivity increased by treating the three-component films with methanol, without deteriorating the Seebeck coefficient. This resulted in a higher thermoelectric power factor of 58.6 μW·m^−1^·K^−2^. This design can contribute toward developing a new field of organic thermoelectric materials with facile flexibility and long lifetime.

Wu et al. [133] proposed a novel and exciting strategy for fabricating an n-type CNT by diethylenetriamine (DETA) doping and the subsequent CaH_2_ treatment of pure CNT. DETA is an amine-contained small organic molecule with lone electron pairs. The DETA doping of the SWCNT enriched the electron density on the SWCNT surfaces, and thus achieved the conversion between the p-type and n-type. Moreover, the reduction/doping mechanism of CaH_2_ facilitated the conversion. In brief, the unstable CaH_2_ initially ionized into 2H^−^ and Ca^+2^, and then combined with 2H^−^ to form H_2_ accompanied with the donation of electrons to the CNT. The DETA-CaH_2_-CNT had a Seebeck coefficient of 41.0 μV·K^−1^ and electrical conductivity of 165.0 S·cm^−1^. Then, the research group fabricated multilayered alternating stacked thermoelectric modules, and the modulus containing 14 couples exhibited an open circuit voltage of 62 mV at the 55 K temperature gradients, and an open circuit voltage of 125 mV at the 110 K temperature gradients. The strategy developed in this study provided a new approach toward obtaining n-type organic thermoelectric materials and devices. Additionally, these organic thermoelectric modulus exhibited the best thermoelectric performance observed thus far.

The p-type or n-type thermoelectric materials can be prepared by doping the CNT with conjugated polyelectrolytes (CPEs), as has been demonstrated by Mai et al. [144]. Their results revealed that the selectivity of charge-transfer doping depends on the polarity of the pendant ionic functionalities of the CPE. Additionally, n-type and p-type thermoelectric materials can be fabricated by doping the cationic CPEs and anionic CPEs to the CNT. High electrical conductivity was obtained by employing the self-doped CPE-Na to disperse the CNT, owing to the minimized inter-CNT connection distances. The cationic CPE-doped CNT had a negative Seebeck coefficient, which is characteristic of the n-type doping approach. This study provided a new strategy, owing to the synthetic versatility of the cationic CPE in the fabrication of n-type CNT thermoelectric materials using solution-processing methods such as drop-casting, spin-coating, and inject-printing.

Kim et al. [145] were the first to employ a wet-spinning process in the fabrication of thermoelectric SWCNT/PEDOT:PSS composite fibers with a SWCNT concentration of 10%–50% in a methanol/coagulation system. The prepared samples were immersed in hydrazine solutions and PEI-infiltration to realize the p-type to n-type transition. The prepared thermoelectric composite fibers had a p-type power factor of 83.2 μW·m^−1^·K^−2^, and an n-type power factor of 113 μW·m^−1^·K^−2^. Subsequently, the organic thermoelectric device was assembled using 12 pairs of p-type and n-type fibers. The results with regard to the voltage-output current and power-output current were obtained by applying a temperature difference of 10 K. A maximum output power of 430 nW was obtained, which confirms the material’s superior capability of generating electric power.

A very large power factor of 1050 μW·m^−1^·K^−2^ for SWCNT/PEDOT composites treated by tetrakis(dimethylamino)ethylene (TDAE) has been obtained by Wang et al. [146]. It is noteworthy that after treatment with TDAE, the SWCNT and PEDOT were converted to the n-type, owing to the change of the carrier majority, while the Seebeck coefficient of these composites ranged from −0.84 to −4.3 mV·K^−1^. The large Seebeck coefficient is attributed to the partially percolated SWCNT networks and the distinctly decreased electron concentrations. Unlike other studies, the electrical conductivity of the composites considered in this study slowly increased with the SWCNT loading, as a result of being treated with TDAE. Pristine SWCNT has an extremely high thermal conductivity, which is higher than 1000 W·m^−1^·K^−1^. However, the high thermal conductivity of the SWCNT can significantly decrease by the thermal resistances at the inter-SWCNT and polymer junctions, owing to the heterojunction. At room temperature, the thermal conductivity of the prepared materials was 0.67 W·m^−1^·K^−1^. Additionally, the highest figure of merit was calculated as 0.5 at 300 K, which outperforms that of any other organic n-type materials reported thus far.

### 3.2. Graphite and Its Derivate-Based Organic Thermoelectric Materials

#### 3.2.1. Graphene-Based Organic Thermoelectric Materials

As a novel and burgeoning two-dimensional crystalline carbon material, graphene is provided with remarkably high electrical conductivity and satisfactory thermoelectric performance [147,148,149,150,151]. In this section, recently developed organic thermoelectric materials containing graphene oxide, reduced graphene oxide, graphene, and graphene (graphite) nanoplatelets are discussed along with their advanced fabrication methods, related achievements, and potential applications. A summary of thermoelectric results of graphene- or graphite-based organic thermoelectric materials is presented in Table 3.

Yoo et al. [155] were the first research group to prepare PEDOT:PSS/graphene thermoelectric composites using an in-situ polymerization method. The graphene particles were charged into the PSS solution for dispersion, and EDOT monomer was added subsequently to realize the polymerization. With the incorporation of 3 wt % graphene, an electrical conductivity of 637 S cm^−1^ was obtained, and was 41% higher than that of the pure PEDOT:PSS film. The Seebeck coefficient of the graphene-incorporated composites produced results similar to those obtained for a pristine PEDOT:PSS film. The highest thermoelectric power factor of 45.7 μW·m^−1^·K^−2^, which is 93% higher than that of pristine PEDOT:PSS film, was obtained for the PEDOT:PSS composites with a graphene content of 3 wt %.

Xiong et al. [159] prepared PEDOT:PSS nanocomposites with high electrical conductively and achieved a remarkable increase of the thermoelectric power factor using graphene doping and hydrazine treatment. It is noteworthy that the electrical conductivity of films containing 3 wt % graphene decreased from 1298 to 783 S cm^−1^ as the hydrazine concentration increased, because the hydrazine resulted in neutral PEDOT. The Seebeck coefficient first increased with the increase of hydrazine concentration, and then decreased. A maximum thermoelectric power factor of 53.3 μW·m^−1^·K^−2^ was obtained for a hydrazine concentration of 0.005 M, which is almost two times higher than that of neat PEDOT:PSS composites. With an assumed thermal conductivity of 0.3 W·m^−1^·K^−1^, the ZT was calculated as 0.05 at room temperature, which outperforms the value obtained by most other studies.

Wang et al. [160] prepared PANi/3D tubular graphene composites using the in situ polymerization method, and achieved high electrical conductivity and low thermal conductivity owing to the good dispersion and enhanced interface amounts between the PANi and the 3D graphene. The electrical conductivity of the PANi/graphene composites decreased as the graphene content increased. The Seebeck coefficient of the PANi/graphene composites containing 55–85 wt % graphene was much larger than that of pure PANi and graphene. The unique structure of the PANi/graphene composites could enhance the energy filtering effect and, thus, increase the thermoelectric performance of the materials. The PANi/graphene composites with a graphene content of 67 wt % had the highest Seebeck coefficient of 39 μV·K^−1^. The power factor increased with the 3D graphene content, and then decreased. A maximum thermoelectric power factor of 13.2 μW·m^−1^·K^−2^ was achieved for the PANi/graphene composites with a graphene content of 55 wt %.

Lin et al. [161] fabricated diamino-modified graphene/PANi composites capable of practical high thermoelectric performance. In brief, p-phenediamino was used to modify the graphene (PDG) via a facile one-step chemical method. The PDG were copolymerized with the aniline monomers to fabricate PDG/PANi thermoelectric composites with unique semi-interpenetrating networks. This network cannot only create more pathways to realize a smooth carrier transport, but also uses less graphene to enhance the overall thermoelectric properties, such as the Seebeck coefficient and electrical conductivity. The PANi/PDG composites with a modified graphene content of 1 wt % exhibited an electrical conductivity of 37 S/cm, which is seven times higher than that of pristine PANi/graphene composites. The enhanced electrical conductivity is attributed to the enhanced conjugated cross-links formed between the PANi and the PDG, which can facilitate the carrier transport. The Seebeck coefficient decreased as the filler concentration increased, while the thermal conductivity increased. The highest ZT value of 0.74 was calculated for the PANi/PDG samples containing only 3 wt% modified graphene.

#### 3.2.2. Graphite- and Expanded-Graphite-Based Organic Thermoelectric Materials

As a crystalline allotrope of carbon, graphite has been widely used in thermochemistry owing to its high stability and conductivity. Graphite has a layered structure, and the atoms are arranged in a honeycomb lattice with plane distances of 0.335 nm. Owing to the excellent properties of graphene, the studies investigating graphite have been covered. However, graphite still acts as an ideal filler owing to its low-cost and intrinsic properties.

Expanded graphite is derived from expandable graphite, which is an artificially synthesized intercalation compound of graphite that will expand under heat treatment. Expanded graphite is a 3D material with high electrical conductivity and superior thermal conductivity, and is typically used in the development of thermal management applications.

Du et al. [162] prepared PEDOT/graphite composites using an in-situ oxidative polymerization process. The thermoelectric performances of the prepared nanocomposites were measured under different temperatures. The electrical conductivity sharply increased from 3.6 to 80.1 S/cm, while the Seebeck coefficient was similar within the value range of 12 to 15.1 μV·K^−1^ when the graphite loading increased from 0–37.2 wt %. With a graphite loading of 37.2 wt % at 380 K, a power factor of 3.2 μW·m^−1^·K^−2^ was obtained, and was 30 times higher than that of neat PEDOT.

Pan et al. [163] used a simple ultrasonic mixing method to prepare PANi/expanded graphite composites with enhanced thermoelectric performance. The prepared composites had an interesting structure: the PANi was intercalated into the expanded graphite sheets, thus forming a sandwich structure. The electrical conductivity and Seebeck coefficient of the prepared composites increased with the increase of the expanded graphite concentration. A maximum power factor of 2.43 μW·m^−1^·K^−2^ was obtained owing to the high electrical conductivity and Seebeck coefficient.

According to the abovementioned results, it can be concluded that the thermoelectric performance of graphite- or expanded-graphite-based organic materials is considerably inferior in comparison with that of graphene- or CNT-based organic materials. The inferior thermoelectric performance has restricted the commercialization of these materials and research on their practical applications.

### 3.3. Ternary Thermoelectric Composites

Ternary thermoelectric composites, such as the PEDOT:PSS/rGO/Te nanowire [164], polypyrrole/graphene/PANi [165], and PEDOT:PSS/SWCNT/Te [166] nanocomposites, have been widely investigated owing to their unique inner structure and the synergistic effect of their different constituents.

Cho et al. [167] fabricated printable thermoelectric composites composed of graphene, PANi, and double-walled CNT (DWNT) by employing a layer-by-layer assembly technique. Conductive PEDOT:PSS was used to stabilize the graphene and DWNT. The three layer-structured thermoelectric materials of PANi/graphene-PEDOT:PSS, PANi/DWNT-PEDOT:PSS, and PANi/graphene-PEDOT:PSS/PANi/DWNT-PEDOT:PSS were fabricated, and the electrical conductivity of all samples increased by increasing the number of cycles. The PANi/graphene-PEDOT:PSS/PANi/DWNT-PEDOT:PSS exhibited the highest electrical conductivity of 1885 S cm^−1^ at 80 quadlayers, which indicates an improved interconnected network for electron transfer. The p-type Seebeck coefficient of these films was obtained as a function of cycle numbers. The Seebeck coefficients of both the PANi/DWNT-PEDOT:PSS and the PANi/graphene-PEDOT:PSS composites increased as the layer cycles increased, reaching 92 and 58 μV·K^−1^, respectively. The PANi/graphene-PEDOT:PSS/PANi/DWNT-PEDOT:PSS film exhibited a similar increasing trend with regard to the Seebeck coefficient, and reached to 120 μV·K^−1^ at 80 quadlayers. The high electrical conductivity and excellent Seebeck coefficient of the PANi/graphene-PEDOT:PSS/PANi/DWNT-PEDOT:PSS film led to an exceptionally high thermoelectric power factor of 2710 μWm^−1^K^−2^, which outperforms the value reported for other organic thermoelectric materials at room temperature. This design provides a good opportunity to use waste heat in an environmentally efficient manner.

Li prepared [151] poly(3,4-ethylenedioxythiophene)/graphene/CNT ternary composites with improved thermoelectric performance using in situ polymerization and physical mixing methods. The thermoelectric performances of the untreated and H_2_SO_4_-treated samples were compared. The PEDOT/rGO/SWCNT ternary composites exhibited the highest electrical conductivity in comparison with the neat PEDOT and PEDOT/rGO binary composites. The increased electrical conductivity is attributed to the integrated electrical network, alignment of the PEDOT molecules, and transition of molecular conformation. The acid treatment of these samples resulted in a considerably higher electrical conductivity, owing to the removal of the neutral PSSH molecules. Simultaneously, the ternary structure and acid treatment had a significant effect on the Seebeck coefficient. The power factor of the ternary composites increased with the increase of the CNT content, which is mainly attributed to the significant increase of electrical conductivity. The post-treated PEDOT/rGO/SWCNT ternary composite that contained 10 wt % CNT had a power factor of 9.1 μW·m^−^^1^·K^−^^2^, which is four times higher in comparison with that observed without treatment.

## 4. Conclusions and Prospects

In this review, we briefly introduced the parameters controlling the thermoelectric performance. Additionally, we summarized various recently developed carbon-material-based organic thermoelectric composites. Carbon-based thermoelectric materials are characterized by the decoupling of electrical conductivity and Seebeck coefficient; thus, their thermoelectric performance is promising. The advanced preparation methods and thermoelectric performances of recently-developed thermoelectric materials were presented. These pioneering studies have facilitated the commercialization of organic thermoelectricity.

Although organic thermoelectric materials have various merits, such as low cost, low thermal conductivity, easy-processing, and good flexibility, their thermoelectric efficiency is still very low (approximately 5–10%), which restricts their commercial application. Owing to the demand for green energy from thermoelectric materials, highly efficient thermoelectric materials should be developed in the future. The interactions amongst the Seebeck coefficient, electrical conductivity, and thermal conductivity should be further optimized, in addition to conducting a fundamental investigation into the effect of structure-property relationships on thermoelectric performance.

## Figures and Tables

**Figure 1 polymers-11-00167-f001:**
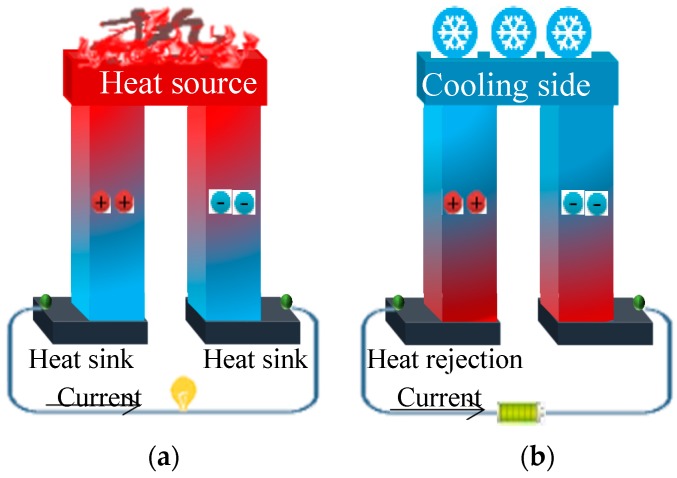
(**a**) How heat can be converted to electricity (Seebeck coefficient) using thermoelectric materials; and (**b**) how electrical power can be transformed to cooling or heating (Peltier effect).

**Figure 2 polymers-11-00167-f002:**
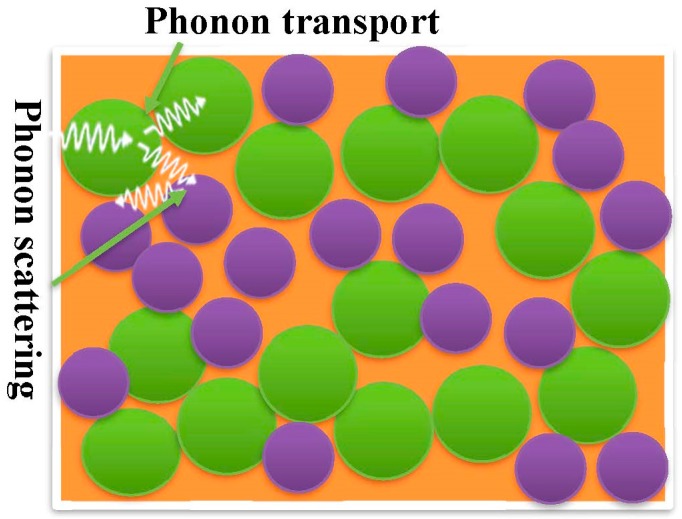
Thermal conduction in amorphous polymers.

**Figure 3 polymers-11-00167-f003:**
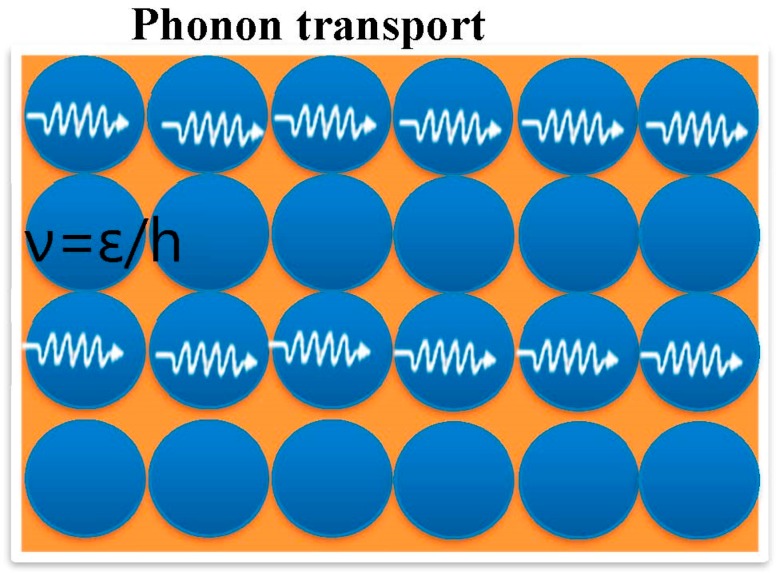
Thermal conduction in ideal crystalline materials.

**Figure 4 polymers-11-00167-f004:**
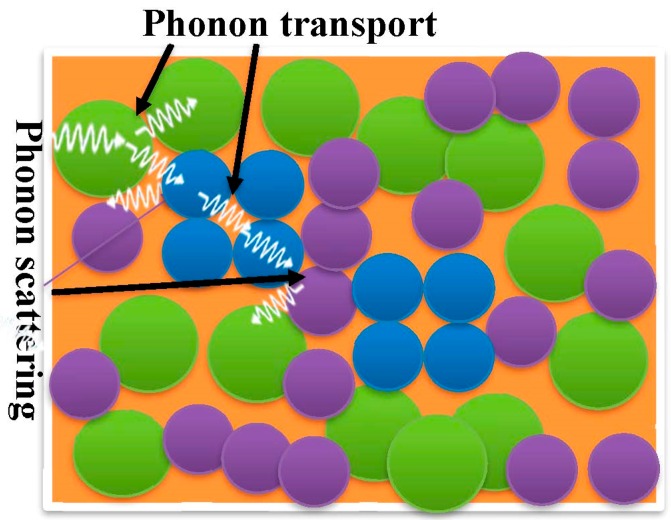
Thermal conductivity in crystalline filler-based polymer composites with discontinuous filler network.

**Table 1 polymers-11-00167-t001:** Chemical structure of various typical thermoelectric polymers.

Materials	Chemical Structure
Polyaniline	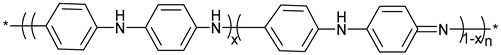
Polypyrrole	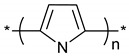
Polythiophene	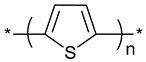
Polyacetylene	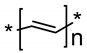
Poly(3,4-ethylenedioxythiophene)	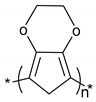
Polyphenylenevinylene	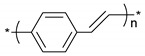
Poly(3-methylthiophene)	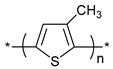
Poly(2,7-Carbazolylenevinylene)	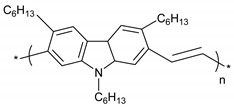
Poly(3-octylthiophene)	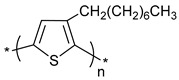
Poly(3-hexylthiophene)	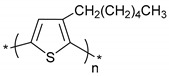
Poly(vinylidene fluoride)	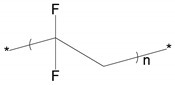

**Table 2 polymers-11-00167-t002:** Summary of thermoelectric results of CNT-based organic thermoelectric materials.

Materials	σ (S/m)	*S* (µV/K)	κ (W/mk)	*PF* (µW/mK^2^)	*ZT*	Temperature (K)
P3HT/CNT [125]	345 ± 88	97 ± 11		325 ± 101		298
PEDOT:PSS/SWCNT [61]	~10^5^	41	0.2–0.4	~160		300
PANI/CNT [126]	~6000	~29		~5		298
Nafion/MCNT [127]	1300	24		0.07	0.001	298
PEDOT:PSS/MWNT [128]	9500	40	0.12	20		298
PEDOT:PSS/DWNT [128]	96,000	70		500		298
PANI/SWCNT [129]	76,900	65	~0.2	176	0.12	298
PEDOT:PSS/CNT [130]	78,000 ± 5100	43.7 ± 3.3		151 ± 34		298
P3HT/SWCNT [131]	~10^5^	~35	~0.19	95 ± 12	>10^−2^	298
PANI/SWCNT [132]	1.25 × 10^4^	40		2 × 10^−5^	0.004	298
Diethylenetriamine/SWCNT [133]	16,500 ± 1000	−41.0 ± 1.5				298
PANI/SWCNT [134]	1.44 × 10^5^	40	0.44	217		298
PEDOT/SWCNT [135]	57,040 ± 1580	17.5		19.00 ± 1.43		298

**Table 3 polymers-11-00167-t003:** Summary of thermoelectric results of graphene- or graphite-based organic thermoelectric materials.

Materials	*σ* (S/m)	*S* (µV/K)	κ (W/mk)	*PF* (µW/mK^2^)	*ZT*	Temperature (K)
PEDOT/rGO [152]	5000	31.8		5.2 ± 0.9		298
PEDOT:PSS/graphene (100:1) [153]	1469	46.9	0.19	3.23	0.00046	300
PEDOT:PSS/graphene (100:2) [153]	3200	59	0.14	11.09	0.021	300
PEDOT:PSS/graphene (100:3) [153]	3170	44.75	0.3	6.34	0.00057	300
PANi/Graphene [154]	5000	30		5.6		298
PEDOT:PSS/Graphene [155]	63,700	26.778		45.67		298
PANI/GNPs [156]	5900	33	13		1.51 × 10^−4^	300
PANi/Graphene pellet [157]	5889	31		5.6		300
PANi/Graphene film [157]	863	41.3		1.47		300
PANi/Graphene [158]	2800	25		2.6	0.000195	453

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
