# Peer review of "Recent Advances in Organic Thermoelectric Materials: Principle Mechanisms and Emerging Carbon-Based Green Energy Materials"

_polymers, 2019, doi:10.3390/polym11010167_

Round 1

Reviewer 1 Report

The paper is a review on “Recent advances in organic Thermoelectric materials...”. The authors explain the basis of the thermoelectric theory and then they inspect and comment recent published work on carbon-based organic thermoelectric materials.

The authors give good explanation about the thermoelectric performance. However, there are some comments that the authors should take into account:

-        In the text they have used “u” instead of “μ; There are some spelling mistakes such as: line 183: conductivit; line  591 stricture.

-        In the text they refer to : Fig. 6 (line612); Fig.6c (line 617); Fig 9 (line 531); Fig 8(b) (line492); Fig 8(a) (line 489). Do they refer to the figure in the reference paper?? It is not clear. As it is written it makes confusion. Can be included the figures in this manuscript?

-        A table summarizing all the found data would help to better visualize and understand the systems commented of this review.

-         The authors should explain the difference, if any, by using DWCNT, MWCNT o SWCNT. They always refer to CNT and there are many differences in thermoelectric behavior among them. They must clarify which kind of CNT are used for every experience.

Author Response

The paper is a review on “Recent advances in organic Thermoelectric materials...”. The authors explain the basis of the thermoelectric theory and then they inspect and comment recent published work on carbon-based organic thermoelectric materials.

The authors give good explanation about the thermoelectric performance. However, there are some comments that the authors should take into account:

Answers: We really appreciated that you took time to review my paper and gave us the valuable comments towards enhancing the quality of this manuscript.

-          In the text they have used “u” instead of “μ”.

-          Answers: Thanks for your corrections, and all of the mistakes have been revised. Please check it.

-           There are some spilling mistakes such as: line 183: conductivit; line  591 stricture.

-          Answers: We are sorry for these mistakes, and we checked through all the manuscrit and revised them. Thank you very much again.

-          In the text they refer to : Fig. 6 (line612); Fig.6c (line 617); Fig 9 (line 531); Fig 8(b) (line492); Fig 8(a) (line 489). Do they refer to the figure in the reference paper?? It is not clear. As it is written it makes confusion. Can be included the figures in this manuscript?

-          Answers: Due to the copyright, these cited figures were deleted. These related figure captions were left. Now we checked all the manuscript and revised them. Thanks again for your valuable corrections.

-          A table summarizing all the found data would help to better visualize and understand the systems commented of this review.

Answers: Thanks for your advise and a  table summarizing all the found data was added in the revised text. Please check it.

-           The authors should explain the difference, if any, by using DWCNT, MWCNT o SWCNT. They always refer to CNT and there are many differences in thermoelectric behavior among them. They must clarify which kind of CNT are used for every experience.

-          Answers: Thanks for your corrections. And we revised them as advised. Thank you for all your valuable coments again. Happy new year!

Reviewer 2 Report

This is a well organized review paper with clear structures including basic introduction to thermoelectric mechanisms, typically carbon-based thermoelectric materials, and prospects. The introduction part is comprehensive for even outside researchers. And the main content on the summary of recent progress on carbon-based materials are well presented.

One suggestion is to use a table to present all listed research work in each section such as CNT, graphene, etc. In the table, authors can list some main features like conductivity, etc. In this way, the readers can have a direct impression on each material and easy to compare among these materials.

Author Response

This is a well-organized review paper with clear structures including basic introduction to thermoelectric mechanisms, typically carbon-based thermoelectric materials, and prospects. The introduction part is comprehensive for even outside researchers. And the main content on the summary of recent progress on carbon-based materials are well presented.

Answers: We really appreciated that you took time to review my paper and gave the valuable comments towards enhancing the quality of this manuscript.

One suggestion is to use a table to present all listed research work in each section such as CNT, graphene, etc. In the table, authors can list some main features like conductivity, etc. In this way, the readers can have a direct impression on each material and easy to compare among these materials.

Answers: Thanks for your suggestions. And we added the tables that summarized the information of recently published papers. Please check it in the manuscript.

Thanks for your comments again. Happy New Year!

Round 2

Reviewer 1 Report

The paper is a contribution presetning recent advances in organic themoelectric materials. Although the authors have not included additional data and have removed figures that could give more information, the paper can be accepted in the present form.